# Present-Day Crustal Velocity Field in Ecuador from cGPS Position Time Series

**DOI:** 10.3390/s23063301

**Published:** 2023-03-21

**Authors:** Alejandro Arias-Gallegos, Mᵃ Jesús Borque-Arancón, Antonio J. Gil-Cruz

**Affiliations:** 1Departamento de Ingeniería Cartográfica, Geodésica y Fotogrametría, Campus de las Lagunillas, Universidad de Jaén, 23071 Jaén, Spain; 2Centro de Estudios Avanzado en Ciencias de la Tierra, Energía y Medio Ambiente (CEACTEMA), Campus de las Lagunillas, Universidad de Jaén, 23071 Jaén, Spain

**Keywords:** GNSS, PPP, time series, Ecuador

## Abstract

The present study analyzes the GNSS time series obtained between the years 2017 and 2022 for the calculation of absolute and residual rates of Ecuador in 10 stations (ABEC, CUEC, ECEC, EPEC, FOEC, GZEC, MUEC, PLEC, RIOP, SEEC, TPC) of the continuous monitoring REGME network. Considering that the latest studies refer to periods 2012–2014 and Ecuador is located in an area of high seismic activity, it is important to update the GNSS rates. The RINEX data were provided by the Military Geographic Institute of Ecuador, the governing institution of geoinformation in that country; for processing, GipsyX scientific software was used with a PPP mode, considering 24 h sessions, and high precision was achieved. For the analysis of time series, the SARI platform was used. The series was modeled using a least-squares adjustment, which delivered the velocities for each station in the three local topocentric components. The results were contrasted with other studies, obtaining interesting conclusions as the presence of abnormal post-seismic rates stands out due to the high rate of seismic occurrence in Ecuador, and reaffirms the idea of a constant update of velocities for the Ecuadorian territory and the inclusion of the stochastic factor in the analysis of GNSS time series, since it can affect the ability to obtain the final GNSS velocities.

## 1. Introduction

As mentioned in [1], the use of GNSS is an important tool that allows for us to understand the behavior of tectonic plates with high precision, providing important information to researchers about the movement of a particular area. In the same way, in this work, it is shown how the GNSS technique could quantify crustal deformation over sites extending east–west from the Caucasus Mountains to the Adriatic Sea and north–south from the southern edge of the Eurasian plate to the northern edge of the African plate. It must be taken into account that the amount of input information to be analyzed also defines the quality of the results as a process method; in this way, it is recommended to consider time series greater than 8 years, as observed in works such as [1,2], where the data of 9 and 16 years are used, respectively. However, the present work tries to model the velocity field of the Ecuadorian crust after a strong seismic event that forced a change in the geodetic reference frame—said event, of which occurred in 2016, limited the analysis to the selected period (2017–2022). Both papers mention a double difference in phases together with precise products to obtain the GNSS positions; however, we worked with precise products and Precise Point Positioning (PPP). Due to this, the method has shown a similar level of accuracy with less computational effort.

Ecuador is a country in South America characterized by the distribution of the territory in the northern and southern hemispheres. The Andes Mountain range crosses it transversally, which therefore defines the country as a territory with high levels of geological activity (Figure 1). Its high deformation and seismic activity rate are also due to the influence of the Nazca and South American plates [3]. On the other hand, the location of the Ecuadorian coast is known for presenting the biggest seismic potential on Earth [4], which has drawn the attention of several researchers who have developed numerous projects and studies. The state-of-the-art leads us to observe projects from 1991, such as the case of the CASA project between Costa Rica, Panama, Colombia, and Ecuador, to a doctoral thesis presented by Luna in 2017 where the velocity field of Ecuador is calculated with data between 2008 and 2014. The aforementioned allows for us to understand that there has been no update of this information between the years 2016 and 2022 (the year in which this research was written); considering that the rates in the tectonic plates are affected for various reasons but mainly by the influence of earthquakes (during and after the event) [5], continuous control of this phenomenon through the analysis of GNSS time series is important. This has been demonstrated in works such as [6,7]. The results allow for us to understand how complex tectonic activity is in South America, specifically Ecuador.

As mentioned at the beginning of this section, the main objective of this research is to model the current velocity field of Ecuador by analyzing the GNSS time series after the most important seismic event of the last 20 years in Ecuador. Considering that a strong earthquake could give important seismic displacements in areas closer to the epicenter [8], we can say that there is an important variation of velocities in the crust. Due to the aforementioned, this research becomes the first modeling of the velocity field of Ecuador after the change in the geodesic reference frame of the country—we started from the selection of the most stable GNSS permanent stations on the territory; then, time series were obtained with subsequent data from the said earthquake to calculate the velocities. With the application of Euler’s pole, it was possible to eliminate the influence of the movement of the plate to obtain the deformations of the Earth’s crust and finally compare the results with previous studies.

## 2. Tectonic Setting

The catalog of seismic events of the Geophysical Institute of the National Polytechnic University of Ecuador records a total of 2833 seismic events greater than 4 Mw from 2012 to the date this study was written (Figure 2). The complex seismic activity of Ecuador is mainly due to the influence of the Nazca oceanic plate with the South American continental plate [9]. This convergence of plates is known as the western edge of South America and its influence is evidenced by the presence of different levels of deformation on the Ecuadorian continental crust, including the formation of chains of high hills approximately 760 m above sea level [10]. It is also known for its great seismic potential, which is considered as the largest on the planet from a seismological point of view [4]. Here, earthquakes originate at different depths and with high magnitudes that can cause major catastrophes, such as the earthquake that occurred in Ecuador on 16 April 2016 in Pedernales, a province of Manabí, which generated impacts throughout the Ecuadorian territory. The South American Plate in turn is divided into two blocks—to the north, the Norandino block (BNA), and to the south, the South American plate proper (SOAM) [3,11,12]. Studies maintain that there is no well-defined guideline that divides these blocks [13]; however, it can be modeled with a Benioff plane in the coastal region, Santa Elena peninsula, with a structural dip between 4° and 5°, at a depth between 40 and 70 km, and a sudden change in the Andean region with a tilted plane between 17° and 28° [9,10]. Figure 3 shows the composition of the blocks and the convergence of the tectonic plates in the Ecuadorian territory.

## 3. Methodology

The methodology is an adaptation of works carried out by Łyszkowicz 2021 [16], Taissia 2020 [17], and Savchuk and Khoptar 2018 [18] for scientific processing with GipsyX software, and an adaptation of the methodology for time-series analysis by Luna et al., 2017 [19]. It is important to mention the principal steps followed: preprocessing when the quality control was carried out, the scientific process using GipsyX software, and the time-series analysis where the SARI platform was used.

### 3.1. GNSS Data Processing

Daily RINEX files belonging to 10 stations of the continuous monitoring network of Ecuador (REGME) were used for periods between 2017 and 2022 (Figure 4). The REGME was created and administered by the Military Geographic Institute of Ecuador (IGM).

This geodetic network has double frequency geodetic antennas linked to SIRGAS (Geodetic Reference System to America), continuously catching GNSS data (GPS and GLONASS) for 365 days every year. This network establishes the Geodetic Reference frame in Ecuador (SIRGAS-Ecuador) with a similar definition to ITRF.

SIRGAS is a union of government agencies, universities, and research centers focused on geodesy and cartography that has as an objective. SIRGAS has the generation and handling of the regional geocentric reference frame, a vertical reference frame, a gravimetric geoidal model, and a continental gravimetric network. SIRGAS provides products such as semi-free weekly solutions from SIRGAS-CON stations, weekly coordinates of the SIRGAS-CON stations’ multi-year solutions (positions and rates) from SIRGAS-CON, the VEMOS velocity model, and tropospheric delays.

To select the 10 stations, their distribution over the Ecuadorian territory, antenna/receiver changes, and the quality of the information contained in each RINEX (Table 1) were taken as a criterion, for which the quality control module in Teqc software was used, with which the RSM of the multipath effect for L1 (mp1) and L2 (mp2) was evaluated. The results were compared with the average quality of the IGS stations where the values of mp1 and mp2 must be less than 0.3 [20]. For the processing of RINEX files, the GipsyX scientific software [21] was used, with a processing routine in PPP mode, together with precise products following [22], such as ephemeris, Earth orientation parameters, and ocean tide models, as recommended by the IERS 2010 conventions [23] in the reference frame IGS14.

The analysis software from JPL (Jet Propulsion Laboratory, a federally funded research and development center managed for NASA by Caltech, California Institute of Technology, USA), used a Kalman Filter to estimate combined geodetic parameters. The gd2e.py program estimates the GNSS station position every day from daily RINEX data, downloading the precise products needed from the JPL server while executing the gd2e.py module. This processing routine is commonly used to estimate positions and velocities in various investigations in the field of GNSS [24].

Once the position data were processed, it was necessary to transform from geocentric global coordinates X, Y, and Z to local topocentric coordinates north, east, and up to carry out an analysis of the horizontal and vertical components in separate ways. For that, a routine was used based on the difference between the geocentric coordinates and the origin coordinates where the first data in the time series were used as an origin; then, multiplication with the rotation matrix of geodetic components of the origin coordinates was carried out. In the same way, the standard deviations were switched to the local system using the general formula of uncertainty propagations.

### 3.2. GPS Position Time Series

All time-series analyses were performed using the SARI platform developed by the Observatoire Midi-Pyrénées, which tool, designed in the R environment, allows for us to work with time series, mainly GNSS, adjusting series to a mathematical model. At the same time, it allows for a spectral analysis. For this, the coordinates obtained from the GNSS processing in GipsyX were transformed from Cartesian geocentric (X, Y, Z) to local topocentric (e, n, up) with MatLab. In the same way, we proceeded with their respective standard deviations, as specified in the previous section. Local topocentric coordinates were the input for SARI. Time-series analysis is based on an additive decomposition adjusted by least squares, where it is known that a periodic signal has the main frequency and the highest harmonics—this helps to model the signal and separate the deterministic part (trend and seasonality) and the stochastic part (signal noise) [25]. The theoretical model can be summarized according to the following expression:(1)Y=T+E+R 
where Y is the observed variable, T is the trend, E is the seasonality, and R is the noise or random components [3].

The time analysis starts from the adjustment using a linear regression of the time series to a straight line, which will describe the general behavior of the series (Trend). The trend model is expressed according to Expression (2):(2)Y=Intercept+rate×t
where Intercept represents the ordinate at the origin, rate the velocity, and t the observed instant of time. For estimating the station’s velocity, it is possible to use the daily solutions and full covariances using the least squares method, analyzing every single component (e, n, up) by obtaining a trend in the position’s time series [26]. At this point, it is important to eliminate the outliers that can be considered anomalous values in the time series. This would be complex because this type of series is not stable over time and it is a trend that can increase or decrease; however, to eliminate the outliers, we proceeded as [27]. Outliers were disregarded with an error bar greater than three times the components’ root mean square (RSM) dispersion.

Other types of error to be modeled are the offsets that represent jumps in the series. Those can be caused by natural sources, such as earthquakes, or human sources such as antenna/receiver changes. Offset detection becomes one of the more important steps for a successful time-series analysis because a bad GNSS velocities estimation may be due to a bias generated by a poor offset model. To obtain models about these jumps, SARI software uses an important tool for automatic detection, and then it is necessary to evaluate those detections to obtain a final offset model using a correlation matrix in every single component.

Next, we proceeded to analyze seasonality, which is the oscillations that can be repeated from time to time. In SARI, it is carried out by including the sinusoidal component using Fourier series (S1×sin(2πt), C1×cos(2πt)) in Expression (2), as shown in Expression (3):(3)Y=Intercept+rate×t+S1×sin(2πt)+C1×cos(2πt)

To study the periodicity of the sinusoidal component, the amplitude spectrum is analyzed to quickly evaluate the sinusoidal peaks of the series, based on the Lomb–Scargle periodogram [28] with a normalization that forces the power spectrum’s integral to equal the total variance of the observed series [29] (Figure 5).

The periodogram also allows for us to perform a spectral analysis of the noise (Figure 5); with this, it is possible for us to assign more realistic uncertainty values to the model [30]. For this, it is necessary to find the value of the power index by adjusting a straight line according to the Expression (4), [31]:(4)$$P_{x}\left(f\right)=P_{0}{\left(\frac{f}{f_{0}}\right)}^{v}\;$$
where v is the spectral index, represented by the slope of the fitted line (Figure 5), $P_{0}$ and $f_{0}$ are normalizing constants, and f is the temporal frequency. Depending on the value of v, the noise can be white, pink, or red [32].

In this way, the absolute velocities of the analyzed GNSS stations are obtained. However, to perform an unbiased analysis, it was necessary to transform the velocities obtained in ITRF2014 to a reference frame fixed to the South American plate (SOAM), thus obtaining residual velocities. For this, by using values for the Euler pole proposed by Altamimi et al. [33], the plate movement was eliminated, thus obtaining the rate of deformation of the Ecuadorian crust. The velocity of the plate is given by the cross product vi = Ω × Ri, where Ri is the position of the plate and Ω is the angular velocity vector of the plate.

## 4. Results and Discussion

The results of the temporal analysis allow for a graphical visualization of the series adjustment. An example is presented for the EPEC station, the adjustment of the additive decomposition in trend (Figure 6), trend after eliminating the outliers and offset model (Figure 7), trend plus seasonality, and eliminating the outliers and the offset model (Figure 8).

It is important to notice the difference in how to fit the theoretical model in Figure 6 and Figure 8, where the offset detection and seasonality component make a big difference, allowing for us to obtain a better rate estimation.

Thus, a summary is also obtained where the values of the adjustment made are observed, from which the values of the rates and their corresponding uncertainties are extracted and presented in Table 2.

The uncertainties shown in Table 2 are produced by the spectral properties analysis in each station and component, and give the quality of the velocity estimations. In general, there is not a large difference between the horizontal uncertainties—the average of these was ∼0.13 mm in the east component and ∼0.11 mm in the north component, while the average in the up-uncertainty component was ∼0.27 mm, where we can find an important difference. However, these final uncertainties demonstrate high-precision observations.

Figure 9 shows the velocity field in the studied region.

Comparing Figure 9a and Figure 9b, we can see the South American Plate (SOAM) contribution, while the first one shows a movement in direction principally to the north, and the second figure shows the deformation of the Ecuadorian territory.

The results of this study were contrasted with other studies, obtaining interesting conclusions. Firstly, the tectonic block model proposed by Chunga et al. in [10] has a strong relationship with the vertical velocity map (Figure 9b) since it is possible to easily observe the behavior of the Norandino block (BNA) versus the South block (Figure 10). We easily see the northern block’s downward trend vs. the south block’s upward trend.

On the other hand, the absolute rates were compared with the last study carried out by Luna in [15] in six stations (CUEC, ECEC, EPEC, GZEC, RIOP, SEEC) (Table 3). Some differences are evident, especially in the component east, where the greatest difference occurs in the ECEC station with ∼12.98 mm/year. However, the difference in the rest of the stations is around ∼4 mm/year. The differences in the north and up components are much smaller. The largest difference in the north component is at the RIOP station with ∼4 mm/year, and the rest is around ∼1 mm/year; something similar occurs with the up component, where the largest difference is at the ECEC station with ∼4 mm/year, and the rest is around ∼1 mm/year. We need to consider that the stations with the most significant differences are also stations where Luna [3] used data only for 1 or 2 years to make the time series—this can provide uncertainty at the moment regarding a comparison of the results.

To explain these differences, we must consider as well—among the most important factors—the seismic event that occurred in Ecuador on 16 April 2016 (one year after the studies from Luna and one year before the studies of the present work). This event of magnitude 7.8 MW, unfortunately, caused several human losses and also affected the REGME GNSS stations, for which the Military Geographic Institute of Ecuador carried out a study on the affected stations [34], of which there are three coincident with the stations analyzed in Table 3. There is a clear inversely proportional correlation between the largest differences in the previous table with the distance in km to the epicenter of the seismic event; thus, the station with the smallest difference is RIOP with ∼2.2 mm/year, a distance to the epicenter of 270 km. The next station compared was EPEC with a difference of ∼7.38 mm/year at a distance of 195 km from the epicenter; finally, we found ECEC with a difference of ∼12.98 mm/year to be the closest to the epicenter with 94 km. This could be due to the so-called post-seismic anomalous velocities that are evidenced after strong earthquakes affecting the velocities of the Earth’s crust [35].

When analyzing the north component, which is seen in Figure 9a, we can observe that it has greater influence in the direction of the absolute rates vector. It can also be seen how the velocity vectors are considerably higher in the northern part of the Ecuadorian territory, belonging to the Norandino block. Observing the absolute and residual velocities (Figure 9a and Figure 9c, respectively), what has been discussed is evident—the velocities can reach up to ∼14 mm/year (for absolute velocities), as seen in the north block ECEC station, and in the south block, they reach a maximum of ∼8 mm/year, as observed in the absolute velocities of the CUEC station.

## 5. Conclusions

The PPP technique and the 24 h observation data set were used, following the recommendations of the IERS convections with precise products (antenna models, ephemeris, watches, ocean tides, etc.) To guarantee results with high levels of quality, the temporal series, despite having only 6 years of information, allowed for the modeling of the Ecuadorian velocities’ crust after the 2016 earthquake without any problem.

The temporal series analysis is the basis for the calculation of GNSS velocities; to achieve a reasonable estimate, the stationary processes guarantee a handy tool that, when adapted to the temporary series type, represents the GNSS observations better than other statistical methods, allowing for the elimination of outliers. In the same way, moderating the time jumps produced by offsets allows for the additive decomposition to be free of biases. It is essential to mention that the method selected to eliminate anomalous data and model offsets will depend on the characteristics of the time series.

The influence of noise is an important step in this work, especially for offset modeling. The noise behavior can be seen as a jump with a short dimension. This could be a bias in the compensation model and, of course, in the estimation of velocities. The influence of including the noise analysis on the time series can be seen mainly in the calculation of the rate accuracies; thus, including this analysis allows for us to estimate more realistic accuracies. Additionally, the type of noise can give us an idea of the origin of the error. According to [1], white noise is defined as a random signal with samples without temporal correlation, its main characteristic is that it is independent of the frequency and is related with errors in measurements or errors caused by hardware; with this noise, the uncertainties of the time series can be better estimated, although it lacks geodetic information. On the other hand, pink noise or flicker is the most common in GNSS signals—it has characteristics of white noise and random walk or red noise. Finally, the red noise or random walk allows for us to establish suspicions of instability caused by monumentation.

Summing up, updating GNSS velocities is of utmost importance, especially in areas of active tectonics. It should be considered that after strong earthquakes, there is a relaxation period, and post-seismic anomalous velocities are evident that can drastically change the velocity field. Longer time series from new cGPS data will probably improve the estimations of the velocity field and will allow for us to confirm these results.

As future work, we can add that once the velocity field is obtained, it would be possible to choose—in order to generate a velocity model from the results of this study to obtain rates at the non-sampled sites—a geostatistical method such as Kriging or least squares that adjusts to the movements of the Ecuadorian Earth’s crust.

## Figures and Tables

**Figure 1 sensors-23-03301-f001:**
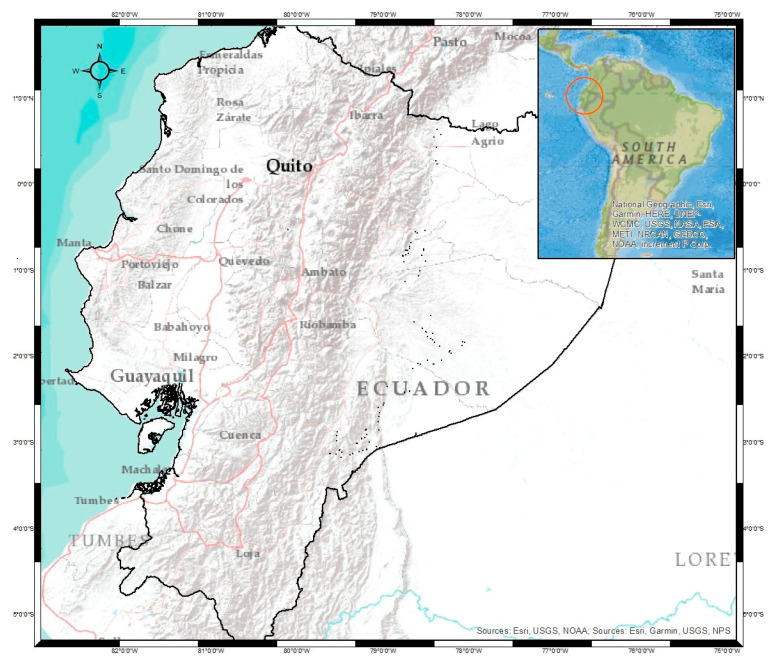
The study area (Ecuador) is localized on South America’s west coast, making it part of the so-called Pacific ring of fire. The relief is characterized by the presence of the Andes Mountain range that crosses from north to south, which generates an important source of earthquakes due to the overpressure of the magmatic fluids of the volcanoes present in this mountain range. These characteristics make Ecuador a territory with high levels of geoactivity.

**Figure 2 sensors-23-03301-f002:**
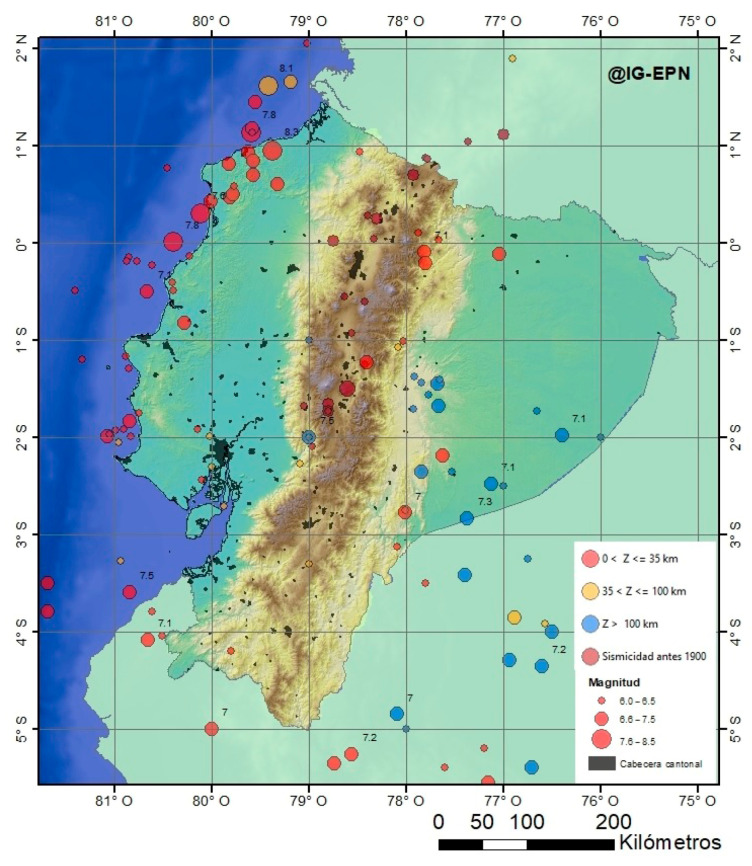
Seismicity of Ecuador with modification in recent years from [14]. The Ecuadorian Institute of Geophysics shows, as one of its products, the map with the geographic distribution of the last earthquakes in Ecuador greater than 6 MW. In this map, it can be seen that the seismic events with the greatest magnitude are concentrated on the Ecuadorian coast.

**Figure 3 sensors-23-03301-f003:**
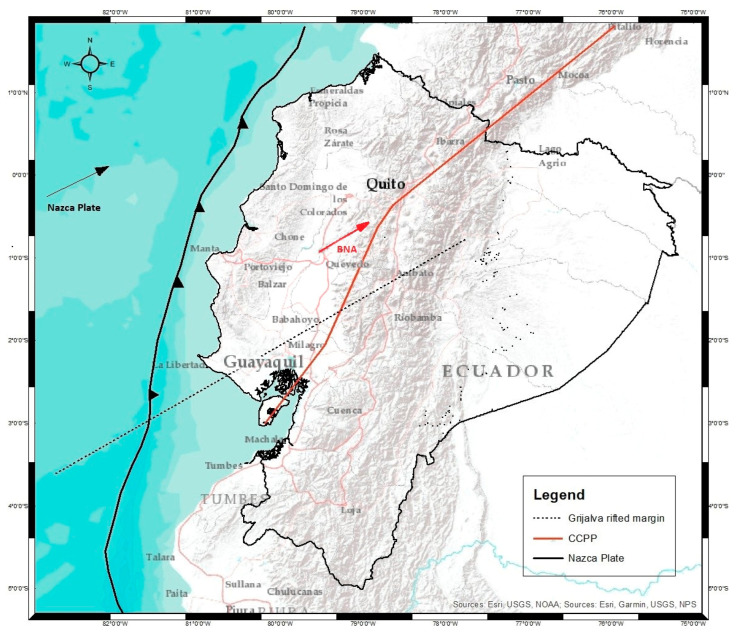
Diagram of the geodynamics of plates in the Ecuadorian territory. The thick black line shows the movement of the Nazca plate concerning the South American plate, and the continuous red line represents the division of the blocks of the South American SOAM plate. This is also represented by the movement of these blocks due to the convexity of the margin (black line with notches). We can observe how the Nazca plate converges—the northern Andean block moves along the right lateral displacement faults or reverse faults. The CCPP fault system continues in Colombia as the Algeciras fault (thick red line). Modification from [15].

**Figure 4 sensors-23-03301-f004:**
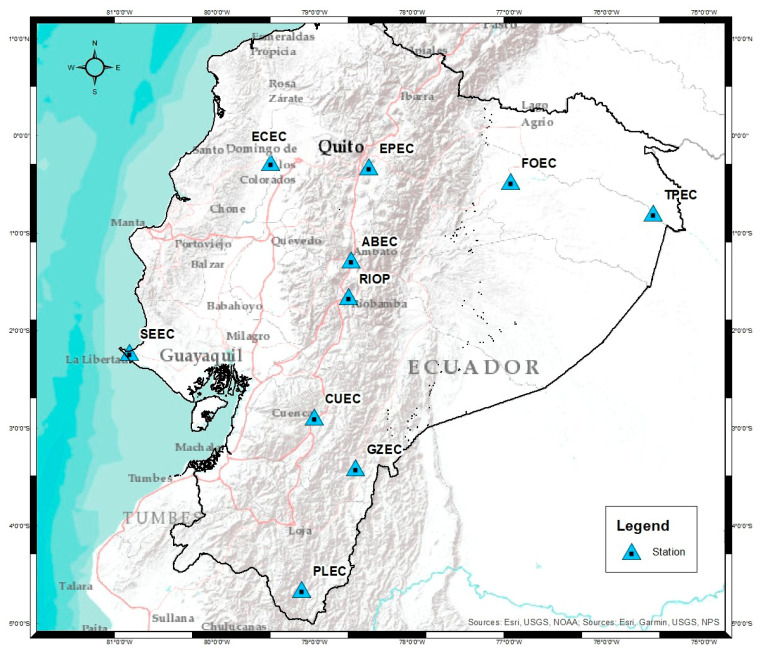
The location of the stations analyzed on the Ecuadorian territory. Each station is part of the REGME (The Ecuadorian GNSS permanent network). It was sought to cover the entire Ecuadorian territory with the selected stations as long as their data met optimal quality levels for temporal analysis. As can be seen on the map, there are 2 stations in the coastal region, 6 stations along the Andes, and 2 stations in the Ecuadorian Amazon.

**Figure 5 sensors-23-03301-f005:**
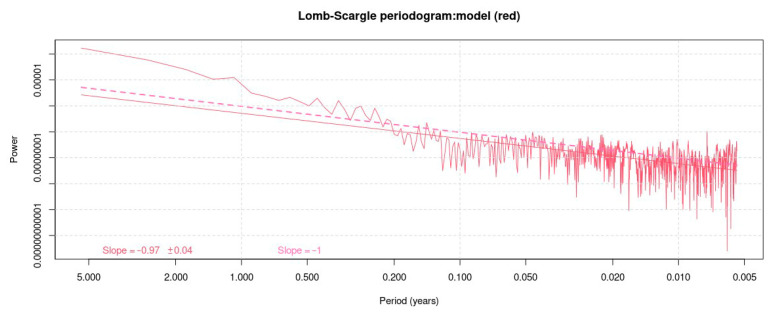
The Lomb–Scargle periodogram for the east component of the EPEC station; the slope of a flicker or pink noise is shown in a dashed pink line, with its value in the lower left.

**Figure 6 sensors-23-03301-f006:**
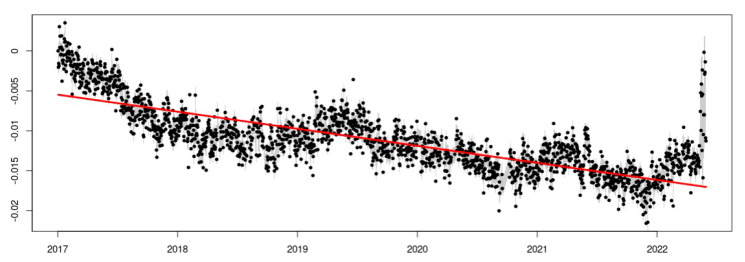
Trend fit for the eastern component of the EPEC station time series. The time series is adjusted with a simple straight line describing a decreasing trend.

**Figure 7 sensors-23-03301-f007:**
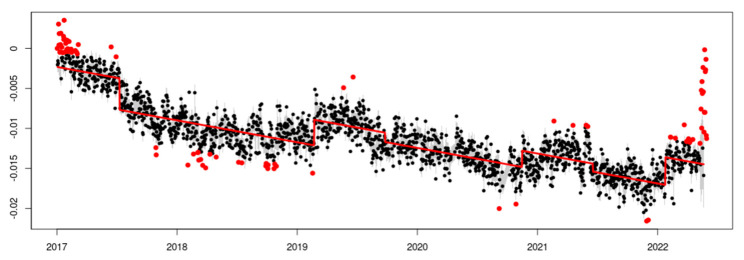
The eliminating outliers and definition of offsets in the time series of the EPEC station. The outliers are in the color red and were not taken into account in the next steps. The offsets are presented in seven sections through the time series.

**Figure 8 sensors-23-03301-f008:**
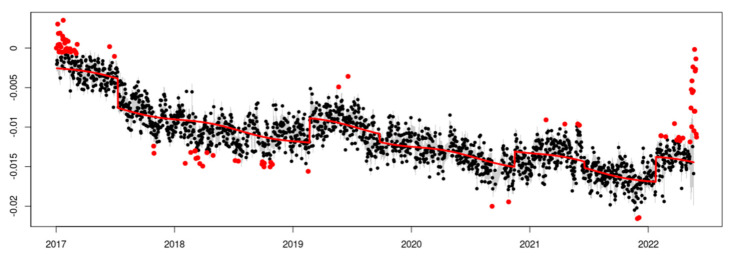
Final adjustment with seasonality in the time series of the EPEC station. The time series fitted is with sinusoidal components.

**Figure 9 sensors-23-03301-f009:**
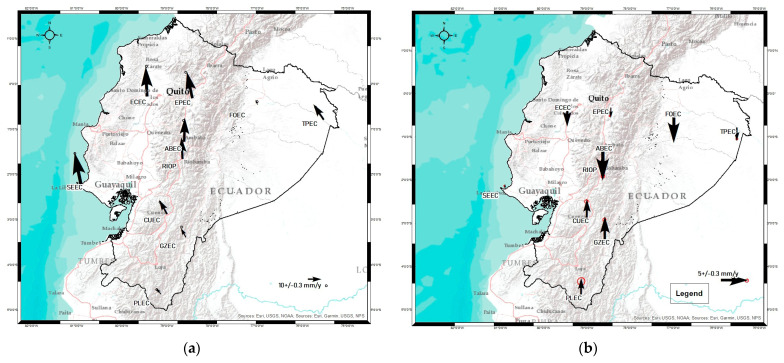
Velocity field from the position time series in the IGS14 reference frame. (**a**) Absolute horizontal rates, (**b**) vertical rates, (**c**) residual horizontal rates. The black arrows indicate the direction and magnitude of the movement, while at the tip of these arrows, there is a red circle that indicates the uncertainty of the movement calculated based on the noise analysis at each station.

**Figure 10 sensors-23-03301-f010:**
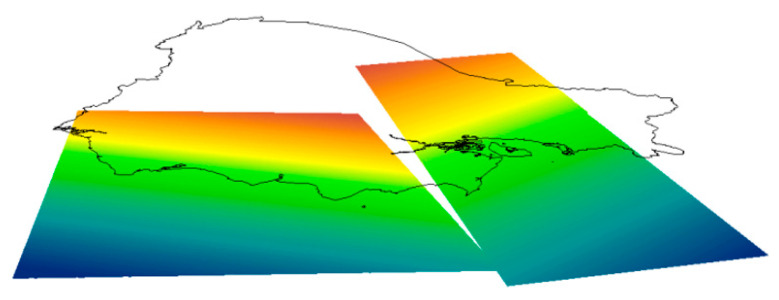
Models of the subduction planes of the Equator. The Benioff plane in the coastal region has a structural immersion of ~5°, between 40 and 70 km deep. A sudden change in dip is delineated in the Andean region, reaching its maximum inclination below the Amazon retro-arc basin, with the plane of inclination between 17° and 28° Taken from [10].

**Table 1 sensors-23-03301-t001:** Summary of stations analyzed showing the number of data years available, the number of antenna/receiver changes, and the start date of the time series.

Stations Analyzed					
Station	Years	Years Range	Antenna/Receptor		
			Receiver Changes	Antenna Changes	Start Date
ABEC	6	2017–2022	2	1	January 2017
CUEC	6	2017–2022	0	0	January 2017
ECEC	6	2017–2022	0	0	January 2017
EPEC	6	2017–2022	0	0	January 2017
FOEC	6	2017–2022	2	2	January 2017
PLEC	6	2017–2022	1	1	October 2017
RIOP	6	2017–2022	0	0	January 2017
SEEC	6	2017–2022	0	0	January 2017
STEC	4	2019–2022	0	0	April 2019
TPEC	5	2018–2022	0	0	January 2018

**Table 2 sensors-23-03301-t002:** Absolute velocities in the north and east components from the GPS position time series in the IGS14 frame. Residual velocities concerning the fixed reference frame of South America plate.

	Geodetic Coordinates (deg.)		Velocity (mm/year)			Uncertainty (mm/year)			Residual Velocity (mm/year)	
Station	Lat. (N)	Long. (E)	ve	vn	vup	σe	σn	σup	ve	vn
ABEC	−1.268	−78.627	0.23	11.19	−5.58	0.06	0.10	0.32	4.36	1.21
CUEC	−2.883	−79.003	−5.12	8.87	2.51	0.18	0.03	0.14	−1.50	1.27
ECEC	−0.272	−79.452	−0.66	13.68	−4.29	0.11	0.09	0.07	3.49	3.86
EPEC	−0.315	−78.446	−2.37	11.24	−0.83	0.13	0.06	0.12	1.18	1.27
FOEC	−0.463	−76.990	−3.43	6.33	−5.60	0.08	0.21	0.22	0.36	−4.03
GZEC	−3.401	−78.581	−4.30	7.57	4.71	0.17	0.11	0.20	−0.74	−2.42
PLEC	−4.651	−79.132	−5.31	6.34	1.87	0.23	0.08	0.38	−1.63	−3.01
RIOP	−1.651	−78.651	−0.57	11.05	0.51	0.05	0.09	0.35	3.40	1.11
SEEC	−2.220	−80.904	−2.95	14.49	0.35	0.13	0.12	0.32	1.04	4.90
TPEC	−0.790	−75.527	−6.01	8.93	−1.03	0.12	0.21	0.53	−2.13	−1.46

**Table 3 sensors-23-03301-t003:** Differences between rates from Luna [3] and this study.

	Last Study (mm/year)				This Study (mm/year)				Difference (mm/year)		
	ve	vn	vup	Time	ve	vn	vup	Time	e	n	up
CUEC	−0.26	7.96	1.6	2008–2014	−5.12	8.87	2.51	2017–2022	4.86	−0.91	−0.91
ECEC	12.32	14.5	−8.95	2012–2014	−0.66	13.68	−4.29	2017–2022	12.98	0.82	−4.66
EPEC	5.01	8.31	−3.16	2013–2014	−2.37	11.24	−0.83	2017–2022	7.38	−2.93	−2.33
GZEC	−0.24	6.33	1.59	2012–2014	−4.30	7.57	4.71	2017–2022	4.06	−1.24	−3.12
RIOP	1.63	6.92	0.69	2008–2014	−0.57	11.05	0.51	2017–2022	2.20	−4.13	0.18
SEEC	9.85	13.14	−1.18	2012–2014	−2.95	14.49	0.35	2017–2022	12.80	−1.35	−1.53

## Data Availability

Not applicable.

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
