# Peer review of "Present-Day Crustal Velocity Field in Ecuador from cGPS Position Time Series"

_sensors, 2023, doi:10.3390/s23063301_

Round 1

Reviewer 1 Report

The objective of this paper is estimating the present-day crustal velocity field from cGPS position time series for the Ecuador region; ten stations were considered for this research, recording data for the 2016-2022 interval.

This is an interesting and well-structured paper, highlighting new aspects of the Ecuador crustal velocity field, based on a geodetic analysis. All necessary sections (Introduction, Tectonic Setting, Methodology, Results and discussion, Conclusions) have been considered. Moreover, the “Methodology” section is divided into sub-sections, providing a detailed analysis. Furthermore, all Figures, Tables and Diagrams are consistent with the analysis provided in the manuscript. However, some changes should be implemented, which will improve the paper. In particular:

Lines 26: The “Introduction” section is brief and should include more details. In particular, two paragraphs should be added; the first paragraph should be placed in the beginning of the section, and it should highlight the cGPS contribution in the crust monitoring. Typical and recent papers, which include this type of information and can be optionally cited, are the following: 1. McClusky, S., Balassanian, S., Barka, A., Demir, C., Ergintav, S., Georgiev, I., Gurkan, O., Hamburger, M., Hurst, K., Kahle, H., Kastens, K., Kekelidze, G., King, R., Kotzev, V., Lenk, O., Mahmoud, S., Mishin, A., Nadariya, M., Ouzounis, A., … Veis, G. (2000). Global Positioning System constraints on plate kinematics and dynamics in the eastern Mediterranean and Caucasus. Journal of Geophysical Research, 105(B3), 5695. https://doi.org/10.1029/1999JB900351, 2. Müller, M. D., Geiger, A., Kahle, H. G., Veis, G., Billiris, H., Paradissis, D., & Felekis, S. (2013). Velocity and deformation fields in the North Aegean domain, Greece, and implications for fault kinematics, derived from GPS data 1993-2009. Tectonophysics, 597–598, 34–49. https://doi.org/10.1016/j.tecto.2012.08.003, 3. Sboras, S., Lazos, I., Mouzakiotis, E., Karastathis, V., Pavlides, S., & Chatzipetros, A. (2020). Fault modelling, seismic sequence evolution and stress transfer scenarios for the July 20, 2017 (M W 6.6) Kos–Gökova Gulf earthquake, SE Aegean. Acta Geophysica, 68(5), 1245–1261. https://doi.org/10.1007/S11600-020-00471-8, 4. Nyst, M., & Thatcher, W. (2004). New constraints on the active tectonic deformation of the Aegean. Journal of Geophysical Research B: Solid Earth, 109(11), 1–23. https://doi.org/10.1029/2003JB002830. The second paragraph should be placed at the end of the “Introduction” section; it should include the paper objectives and the major steps followed to achieve these objectives. Please, apply.

Lines 44-45: A more detailed description should be added in the Figure 1 caption. Moreover, the resolution of Figure 1 is poor and should be increased. Please, apply.

Lines 49-51: The seismic events of the study area are mentioned in this part. Therefore, a map should be added, showing their spatial distribution. Please, apply.

Line 68: In the Figure 2 legend, “CCPP” is included. However, it is not explained in the manuscript or the caption. Please, define.

Lines 95-96: Similarly to Figure 1. See the comments above. Please, apply.

Lines 269-270: Similarly to Figure 1. See the comments above. Please, apply.

Line 331: The “Conclusions” section should be modified. In the current form, it resembles an abstract rather than conclusions. This section should be comprehensive, while the major findings of the paper should be highlighted. Maybe, numbering of the conclusion remarks could be performed. Please, apply.

Reviewer 2 Report

Present-day crustal velocity field of GNSS is studied by many researchers,  and this is a hot topic to the geodesy community.

In this work i did not see anything with new or innovation.

In order to obtain accurate station velocity parameters, a time series of 8-10 years is generally required, and the noise properties should considered.

In this work i did not see much discussion on it, the author should hight-ligh what's new in this work, not just for latest studies refer to periods 2012-2014, and use a new data redo a work.

1. What is the main question addressed by the research? Reply: the figures are with poor quality
2. Do you consider the topic original or relevant in the field? Does it 
address a specific gap in the field? Reply: Yes.  Reply: 3. What does it add to the subject area compared with other published 
material? Reply:More long TS of gnss data is needed.  the noise properties should be involved. 4. What specific improvements should the authors consider regarding the 
methodology? What further controls should be considered? Reply:see 4 5. Are the conclusions consistent with the evidence and arguments presented 
and do they address the main question posed? Reply:Yes, but i did not see to much new compared with the previous studies. 6. Are the references appropriate? Reply:yes.

Round 2

Reviewer 2 Report

the author replied to all my concerns, it can be accepted now.